# Virus-Induced Asthma Exacerbations: SIRT1 Targeted Approach

**DOI:** 10.3390/jcm9082623

**Published:** 2020-08-13

**Authors:** Yosuke Fukuda, Kaho Akimoto, Tetsuya Homma, Jonathan R Baker, Kazuhiro Ito, Peter J Barnes, Hironori Sagara

**Affiliations:** 1Department of Medicine, Division of Respiratory Medicine and Allergology, Showa University School of Medicine, 1-5-8 Shinagawa-ku, Tokyo 142-8555, Japan; yosukefukuda.showa@gmail.com (Y.F.); k_akimoto@med.showa-u.ac.jp (K.A.); sagarah@med.showa-u.ac.jp (H.S.); 2Airway Disease, National Heart and Lung Institute, Imperial College, London SW3 6LY, UK; jonathan.baker@imperial.ac.uk (J.R.B.); k.ito@imperial.ac.uk (K.I.); p.j.barnes@imperial.ac.uk (P.J.B.)

**Keywords:** asthma, exacerbations, virus infection, cellular senescence, SIRT1

## Abstract

The prevalence of asthma has increased worldwide. Asthma exacerbations triggered by upper respiratory tract viral infections remain a major clinical problem and account for hospital admissions and time lost from work. Virus-induced asthma exacerbations cause airway inflammation, resulting in worsening asthma and deterioration in the patients’ quality of life, which may require systemic corticosteroid therapy. Despite recent advances in understanding the cellular and molecular mechanisms underlying asthma exacerbations, current therapeutic modalities are inadequate for complete prevention and treatment of these episodes. The pathological role of cellular senescence, especially that involving the silent information regulator 2 homolog sirtuin (SIRT) protein family, has recently been demonstrated in stable and exacerbated chronic respiratory disease states. This review discusses the role of SIRT1 in the pathogenesis of bronchial asthma. It also discusses the role of SIRT1 in inflammatory cells that play an important role in virus-induced asthma exacerbations. Recent studies have hypothesized that SIRT1 is one of major contributors to cellular senescence. SIRT1 levels decrease in Th2 and non-Th2-related airway inflammation, indicating the role of SIRT1 in several endotypes and phenotypes of asthma. Moreover, several models have demonstrated relationships between viral infection and SIRT1. Therefore, targeting SIRT1 is a novel strategy that may be effective for treating virus-induced asthma exacerbations in the future.

## 1. Introduction

Asthma is the most common chronic respiratory disease, as over 300 million individuals suffer from asthma worldwide [1]. Although the rate of asthma-related mortality has declined for decades due to the advancement in treatment strategies, the prevalence of asthma has gradually increased from 1990 to 2005, and death rates have plateaued in some countries with aging populations [2]. Among all asthma patients, it is thought that about 5–10% of patients have severe refractory asthma. This is due to inaccurate inhalation techniques, poor treatment adherence, and inadequate management of comorbidities [3]. However, even when these factors are excluded, asthma is often still poorly controlled in the population.

Patients still have to grapple with various issues associated with the condition; asthma exacerbation is one of the challenges that requires a more effective solution. Among multiple causes, infectious diseases are the most important cause of asthma exacerbations. Various microorganisms, such as bacteria, fungi, and viruses, can cause acute exacerbations of asthma [4,5,6], and viral infection is the most common trigger.

Human rhinovirus (HRV) is one of the major pathogens of virus-induced asthma exacerbations. In a review of 670 samples of HRV infection detected in the nasal mucosa of infants, HRV caused severe symptoms in winter, and HRV-A and HRV-C caused moderate to severe illnesses [7]. A previous study confirmed elevated expression of interferon (IFN) and Type 2 cytokines analyzed from bronchosorption and nasosorption in asthma patients infected with HRV [8]. In the presence of HRV infection, it was suggested that airway infections of *Streptococcus pneumoniae* and *Moraxella catarrhalis* might increase the risk of experiencing the severity and symptoms of asthma exacerbations [9]. Respiratory syncytial virus (RSV) is also an important pathogen for asthma exacerbations. RSV is known to infect almost all children by the age of 2 years [10], and childhood RSV infection puts adults at risk of developing asthma [11]. In addition, the annual incidence of RSV infection is 3–7% in healthy older adults, and 7.2% of patients hospitalized for asthma have comorbid RSV infection [12]. Thus, it is an essential pathogen for all generations.

Type 2 and non-type 2 airway inflammation are the two major immune phenotypes of asthma [13]. These phenotypes and endotypes are defined by a variety of factors, including genetic predisposition and environmental factors such as antigen exposure, inflammatory biomarkers, tight junctions in the airway epithelium, and viral infections [14,15,16]. In addition to the viral infections themselves, it has been suggested that viral infections can reduce the species, numbers, and diversity of microbiota, so-called dysbiosis [17], indicating that asthma is a very complex disease. Viral infection, which can affect patients with both phenotypes, induces the formation of a wide range of cytokines and chemokines in the airway [18,19,20,21,22]. Eosinophils, T helper-2 (Th2) lymphocytes, type 2 innate lymphoid cells (ILC2), Th17 cells, and neutrophils are involved in epithelial chemokine production in virus-induced asthma exacerbation [18,19,20,21,22] (Figure 1). Inhaled corticosteroids (ICS), which have been used widely for treating asthma over the past few decades, inhibit the expression of inflammatory cytokines in the airway during virus-induced asthma exacerbations, especially Th2-airway inflammation [23]. Although ICS ameliorate asthma exacerbation by limiting neutrophilic and non-Th2 inflammation [24], it is also known that viral infection induces steroid resistance by inducing mainly neutrophilic airway inflammation [25]. Virus-induced asthma exacerbation overburdens healthcare systems, and it elevates the rates of morbidity and mortality [26]. Moreover, a few patients, described as severe, do not respond to current therapies, and few prophylactic strategies are available for treating such refractory cases. Therefore, new therapeutic targets and approaches are the need of the hour.

Cellular senescence is characterized by irreversible cell-cycle arrest and release of inflammatory mediators known as the senescence-associated secreted phenotype (SASP), which can exert paracrine and autocrine effects on naïve cells [27]. The silent information regulator 2 homolog 1 (SIRT1), which is a nicotinamide adenine dinucleotide (NAD+)-dependent class III deacetylase, is one of the essential proteins that regulates aging, metabolism, DNA repair, immunity, and inflammation, and it protects against cellular senescence [28,29].

In recent years, age-related diseases such as heart disease, neurological diseases, cancer, and diabetes have been found to be closely related to SIRT1 [28], and similarly, SIRT1 has garnered considerable attention because of its role in the pathogenesis of asthma [30,31,32]. An SIRT1-targeted treatment strategy may be effective in patients with virus-induced asthma exacerbation, who respond inadequately to existing therapies. In this review, we discuss the function of SIRT1 in inflammatory cells that play a role in virus-induced asthma exacerbations and examine the possibility of using SIRT1 as a target for treating virus-induced asthma exacerbations in the future.

## 2. The Role of SIRT1 in Inflammatory Cells

Activation of SIRT1 induces SASP in T cells through deacetylation of several transcription factors, such as p53, NF-κB, forkhead box O (FOXO)3, PI3K, HIF-1α, and PGC1α [32,33], regulating autophagy, DNA repair, mitochondrial function, and cellular senescence [34]. It was suggested that the mechanism for this was an enhanced glycolysis in helper T cells, which might lead to immune dysfunction [35]. On the other hand, the details of B cells involved in the humoral immune response are not yet well known [36]. In 2003, a screen for mammalian SIRT1 activators identified SIRT1 activators, including resveratrol, piceatannol, and quercetin, called Sirtuin Activating Compounds (STACs). Among them, resveratrol was shown to be the most potent activator of SIRT1 [37]. In a clinical trial of SIRT1 activators in mild to moderate ulcerative colitis, SRT2104, a SIRT1 activator, was well tolerated [38]. Adverse events of SRT2104 was reported to include upper abdominal pain, fatigue, photophobia, diarrhea, and headache [38]. It is considered to be a relatively safe drug to use. On the other hand, selisistat (Ex527), a SIRT1 inhibitor, has been studied in healthy individuals and patients with Huntington’s disease, and has also been shown to be safe [39,40]. The association between SIRT inhibitors and respiratory illness is not well reported. Based on these previous reports, we will discuss about the impact of SIRT1 on virus-induced asthma exacerbations below.

### 2.1. Neutrophils

#### 2.1.1. Neutrophils in Virus-Induced Asthma Exacerbations

Neutrophilic inflammation occurs in the airway during virus-induced asthma exacerbation. CXCL8 is a crucial cytokine of the neutrophilic airway inflammatory process, which is also involved in virus-induced asthma exacerbation [41]. HRV infection increases CXCL8 and IL-1β levels in the nasal lavage fluid of patients with asthma [42]. Smoking is an important factor in the worsening of the disease in asthmatic patients. Stimulating human airway epithelial cells with HRV infection and cigarette smoking extract (CSE) enhances CXCL8 expression [43]. Increased production of CXCL8 is one of the mechanisms of corticosteroid resistance [44,45]. The antimicrobial drug azithromycin may be effective in asthmatics with a predominance of CXCL-8 and other neutrophilic cytokines [46,47]. In a clinical trial of children with RSV-infected bronchitis, azithromycin significantly reduced the expression of CXCL8 in nasal lavage and reduced respiratory symptoms one-year post-use compared to a placebo [46]. Azithromycin may exert its effects by inducing IFN-β and IFN2/3, but the clinical benefits of using antimicrobials, including azithromycin, in patients with asthma are still unclear [48,49].

MMP-9, a type of matrix metalloprotease, has been found to be increased during viral infections [50]. MMP-9 also causes airway remodeling through neutrophilic airway inflammation [50,51]. Airway remodeling is associated with decreased respiratory function and disease severity [52].

Despite the fact that both CXCL8 and MMP-9 are important factors associated with neutrophilic airway inflammation, there are few therapeutic agents for these factors involved in neutrophilic airway inflammation. Furthermore, it may lead to severe virus-induced asthma, and effective treatment strategies are needed.

#### 2.1.2. The Relationship between SIRT1 and Neutrophils in Virus-Induced Asthma Exacerbations

Several reports suggest that SIRT1 regulates neutrophilic airway inflammation related to CXCL8 [53,54,55,56]. CSE-induced CXCL8 elevation in mature mononuclear cells was attenuated by the overexpression of SIRT1 in vitro [55]. These studies showed that activation of NF-κB signaling and deacetylation of FOXO 3a protein as mechanisms by which SIRT1 regulates neutrophilic airway inflammation [55]. A previous study confirmed that FOXO3a expression was upregulated by RSV infection [57]. Thus, SIRT1 activators, including resveratrol, may be effective in targeting CXCL8-induced neutrophilic airway inflammation in virus-induced and steroid-resistant asthma exacerbations [58,59]. Interestingly, in basic experiments with macrophages isolated from bronchoalveolar lavages of COPD patients, resveratrol inhibited the release of nearly all cytokines from alveolar macrophages. In contrast, dexamethasone, a type of systemic corticosteroid commonly used in the treatment of asthma exacerbations, only partially inhibited the release of CXCL8 [60]. These lines of evidence suggest that activation of SIRT1 may lead to suppression of neutrophilic inflammation, possibly through suppression of CXCL8 and may be an effective therapeutic strategy, especially for steroid-resistant virus-induced asthma exacerbations.

Suzuki et al. investigated the relationship between viral infections and MMP-9 expression using human nasal epithelial cells [51]. Notably, they found that the expression of MMP-9, which was enhanced by Poly(I:C), was attenuated by resveratrol. Furthermore, in the presence of the SIRT1 inhibitor splitomicin, Poly(I:C) significantly enhanced the expression of MMP-9 [51]. Another study showed that the increased MMP-9 was attenuated by not only the SIRT1 activator resveratrol, but also by the diabetes drug metformin in a mouse model exposed to UV light [61]. These results indicated that SIRT1 activation could be a novel therapeutic strategy for virus-induced asthma exacerbations by regulating MMP-9 expression and suppressing airway neutrophilic inflammation and remodeling. However, additional studies will be necessary to determine whether MMP-9 is a good biomarker for SIRT1-targeted therapy.

### 2.2. Eosinophils

#### 2.2.1. Eosinophils in Virus-Induced Asthma Exacerbations

Eosinophils play an essential role in virus-induced asthma exacerbation. CCL5 and CCL11, which are chemokines associated with eosinophils, may be upregulated and recruit eosinophils when a virus infects the airway epithelium [62]. Eosinophil cationic protein (ECP), an inflammatory mediator, is released by eosinophils and correlates with airway hyperreactivity [63]. In addition to these, cytokines such as interleukin (IL)-4, IL-5, and IL-13 are thought to be involved in a complex.

Calhoun et al. investigated whether HRV could trigger an allergic response in the airway [64] since it is a significant viral pathogen that exacerbates asthma in adults and children [65]. Bronchoalveolar lavage was performed for both healthy and allergic participants with or without HRV infection, and they concluded that eosinophil recruitment in the airway occurred during or after HRV infection in allergic participants, but not in healthy participants [65]. Kato et al. studied childhood asthma and reported that serum IL-5 and ECP levels were significantly higher in the virus-induced asthma group than those in the control group [66]. They also showed that the profile of those cytokines and chemokines differed with age [67]. Fractional exhaled nitric oxide is a good indicator of eosinophilic airway inflammation [68]. Bjerregaard and colleagues reported that FeNO in virus-induced asthma exacerbation was higher than in the follow-up period [69]. They demonstrated that patients with higher FeNO levels had significantly shorter time to arrive at asthma exacerbation than those patients with lower FeNO [69]. Activation of toll-like receptor 3 (TLR3), a virus receptor within the airway epithelial cells, led to the induction of eosinophil-attracting chemokines (CCL11, eotaxin) in the cellular bases [18,19,20,21,22]. These findings suggested that eosinophilic airway inflammation is an important aspect of virus-induced asthma exacerbation.

#### 2.2.2. The Relationship between SIRT1 and Eosinophils in Virus-Induced Asthma Exacerbations

Several studies have demonstrated the importance of SIRT1 in eosinophilic airway inflammation. Wang et al., using an ovalbumin-induced asthma mouse model, found that SIRT1 was associated with eosinophilic airway inflammation [70]. While they reported that serum SIRT1 levels were increased in OVA-sensitized and challenged mice model, SIRT1 levels were decreased in lung tissue, and more IL-4, IL-5, and IL-13 in BALF were found in the ovalbumin-induced asthma mouse model compared to the controls [70]. They also showed that respiratory function (forced expiratory volume in 1 s/forced vital capacity) was negatively correlated with serum SIRT1 in asthmatic human samples [70]. Based on these results, they believed that the elevated serum SIRT1 levels were due to the release of SIRT1 from the tissues following airway inflammation. In another study, SIRT1 activator (SIRT1720) treatment decreased the eosinophil count and IL-5 and Il-13 levels, but not IL-4 levels in the bronchoalveolar fluid and lung tissue in the ovalbumin-induced asthma mouse model [71]. They also reported that SIRT1 activation significantly inhibited inflammatory cell infiltration in the airways, but it did not significantly affect goblet cell hyperplasia, and they attributed this to the possibility that SIRT1 activation might be inadequate to control airway inflammation. Resveratrol, a known SIRT1 activator, also attenuated IL-5 and IL-13 as well as eosinophil accumulation in ovalbumin-induced allergic rhinitis in mice [72]. This may be attributed to the differential effect of SIRT1 on transcription factors, including GATA3, which is a transcriptional factor that regulates Th2 differentiation and the expression of the T2 cytokines IL-4, IL-5, and IL-13 [73]. SIRT1 is a key regulator of GATA3 via its deacetylation, and in T-lymphocytes from patients with severe asthma, a decrease in SIRT1 has been linked to increased expression of IL-4 via increased GATA3 activation [73]. These findings supported the fact that SIRT1 activation might suppress eosinophilic inflammation in the airway during acute asthma exacerbations.

Controlling eosinophilic inflammation is a key approach for predicting and treating virus-induced asthma exacerbations [68,74,75]. Anti-IL-5 therapies, such as mepolizumab and benralizumab, are effective against eosinophilic airway inflammation and markedly reduce virus-induced asthma exacerbations [76,77]. Although these biologics produce marked effects in patients with refractory asthma, their efficacy was limited in patients with non-Th2-asthma [78,79]. Activation of SIRT1 may facilitate control of eosinophilic inflammation and refractory eosinophilic asthma.

### 2.3. Mast Cells and B Cells

#### 2.3.1. Mast Cells and B Cells in Virus-Induced Asthma Exacerbations

IgE is an important therapeutic target for virus-induced asthma exacerbation, which is mainly produced by plasma cells. The position of omalizumab, which is a humanized anti-IgE monoclonal antibody, has been confirmed as an important therapeutic agent [80]. The PROSE study revealed a lower asthma exacerbation frequency in the omalizumab group than that in the placebo group [81]. In this study, they conducted a subgroup analysis of patients with HRV infection and found a significant increase in IFN-α in the group of patients treated with omalizumab, which may be a protective mechanism for viral-induced asthma exacerbations [81]. Another clinical study showed that omalizumab decreased the frequency and duration of HRV infection in patients with childhood asthma [82]. Despite the efficacy of omalizumab in patients with asthma with viral infection, it was reported that some patients, especially geriatric patients, responded poorly to anti-IgE. This observation may be attributed to immunosenescence, which includes impaired mucociliary clearance, changes within the inflammatory cells in the airway, and decreased antigen response [83]. Hence, there is a demand for other treatment options besides anti-IgE therapy for patients who are unresponsive to anti-IgE therapy.

Lipid profiling is thought to be important for understanding viral infections [84,85]. When the virus reacts with airway epithelium, mast cells are activated in an IgE-dependent or independent manner, and degranulation occurs [86]. This results in the production of lipid mediators such as prostaglandin (PG) and cysteinyl leukotriene (CysLTs), which induce an immediate response in the airways and other target organs [86]. Currently, drugs targeting lipid mediators in asthma are mainly the CySLT_1_ receptor antagonists pranlukast and montelukast [87,88]. It is known that obesity, a factor in refractory asthma, results in steroid resistance due to decreased adipokines. Although leukotriene receptor antagonists are useful in such patients [89], they are still not well controlled. A better understanding of the pathogenesis of refractory asthma by further approaches to lipid mediators is an important issue.

#### 2.3.2. The Relationship between SIRT1 and Mast Cells and B Cells in Virus-Induced Asthma Exacerbations

Resveratrol, a polyphenol found in grapes, berries, red wine, and peanuts [90], can activate SIRT1 [91]. Lee et al. examined the possible anti-inflammatory effects of resveratrol in an ovalbumin-induced asthma mouse model [92]. Resveratrol significantly reduced total IgE and ovalbumin-specific IgE levels and increased serum IgG2a, which is associated with Th1 response, in serum [92]. Moreover, it reduced airway hyperresponsiveness and mucus hypersecretion compared to a placebo [92]. Yet another study found that SIRT1 regulated the pathways, AMP-activated protein kinase (AMPK), and protein tyrosine phosphatase 1B. Modulation of these pathways via resveratrol attenuated signals from the IgE receptor, FcεRI, and inhibited the release of lipid mediators, leukotriene C_4_ (LTC_4_) and PGD2, and the inflammatory cytokines, tumor necrosis factor (TNF)-α and IL-6 [93]. In a mouse model of OVA-induced allergic rhinitis, SIRT1 administration reduced symptoms such as sneezing and nasal rubbing events, and it led to a significant reduction in serum IgE [94]. It is known that when IgE antibodies bind to the antigen, intracellular secretory granules are transported to the cell surface, and chemicals, such as histamine, contained in the granules are released. Previous reports demonstrated that SIRT1 inhibited degranulation [95]. The study noted that inhibition of degranulation by SIRT1 may be mediated through inhibition of the response mechanisms of the phosphorylation of protein kinase C (PKC) isomer, PKCμ and PKCθ. These data suggested that SIRT1 activation may ameliorate IgE-mediated airway inflammation in viral-induced asthma exacerbations, whereas the detailed mechanism by which omalizumab blocks IgE is unclear and requires further study.

Some studies on the association between SIRT1 and lipid mediators have been reported. Tan et al. demonstrated in basic experiments using eosinophils isolated from whole human blood that trans-resveratrol suppressed the expression of LTC_4_ [96]. Another study showed that resveratrol reduced the expression levels of LTC_4_ and PGD in a mouse model of eosinophilic sinusitis [97]. These results might be attributed to inhibition of the phospholipaseA2 (PLA2) and lipoxygenase (LOX) pathways, which play an important role in the arachidonic acid cascade [96,97]. As mentioned above, SIRT1 is closely related to metabolism. It was reported that fisetin and licochalcone, which are polyphenols, improve hepatic lipid metabolism via the SIRT1/AMPK pathway in a mouse model [98,99]. In a clinical trial on asthma and diet, a healthier diet correlated with better asthma control [100]. These evidences suggest that drug treatment and diet modification may be one of the lipid mediator-mediated SIRT targeted treatment strategies for virus-induced asthma exacerbations.

### 2.4. Type 2 Innate Lymphoid Cells (ILC2)

#### 2.4.1. ILC2 in Virus-Induced Asthma Exacerbations

ILCs are a novel type of lymphocyte, which have been recently identified as playing an important role in immune diseases. Th2-type cytokines were initially found to be produced solely by Th2 cells, but recent findings show that ILC2 cells, although less numerous than Th2 cells, are more efficient in producing Th2-type cytokines [101]. Moreover, ILC2 has been shown to play an essential role in virus-induced asthma [102,103]. Studies have confirmed that the expression of the upstream cytokines IL-25, IL-33, and TSLP that regulate ILC2 is enhanced in rhinovirus infections [104,105,106,107]. Current knowledge suggests that airway epithelial damage triggered by viral infections promotes the production of IL-25, IL-33, and thymic stromal lymphopoietin (TSLP), which in turn leads to the activation of ILC2 and exacerbation of asthma [102]. We showed that the late addition of budesonide attenuates the increase in TSLP caused by viral infection [19]. Although Tezepermab, a biologic that targets TSLP for treatment, was reported to reduce the frequency of asthma exacerbations [108], the role of ILCS2, including IL25 and IL-33, is not fully understood, and we need to elucidate the full mechanism of its production and develop new treatment options.

#### 2.4.2. The Relationship between SIRT1 and ILC2 in Virus-Induced Asthma Exacerbations

Little is currently known about the association between airway inflammation, SIRT1, and ILC2. Basic experiments using mouse models of allergic diseases demonstrated that resveratrol inhibited the expression of IL-25, IL-33, and TSLP in airway epithelial cells [109]. Resveratrol also reduces caspase-3, an indicator of apoptosis [109]. In basic experiments using a mouse model of HDM-induced asthma, resveratrol reduced cell apoptosis and suppressed the expression of the γH2AX gene, which is associated with DNA damage [110]. Activation of SIRT1 suppresses epithelial damage, which may inhibit apoptosis and control viral-induced asthma exacerbations. Leptin is one of the adipokines secreted by adipocytes and is involved in increased energy metabolism and appetite suppression via hypothalamic receptors [111]. An increase in SIRT1 led to an increase in the sensitivity of leptin [111]. It was suggested that elevated leptin could cause exacerbation of allergic diseases via ILC2 [112,113]. Zeng et al. tested the relationship between leptin and ILC2 in patients with allergic rhinitis [113]. The results showed that leptin expression correlated with the percentage of ILC2 in peripheral blood mononuclear cells. MAPK signaling and PI3K signaling were thought to be the possible pathways involved in this response [112,113]. It was reported that SIRT1 was inhibited by p38 MAPK and PI3K signaling via micro-RNA (miRNA) 34-a and miRNA570 expression, if the airway epithelial cells were subjected to oxidative stress [114,115]. The possible mechanisms by which SIRT1 regulates allergic airway inflammation through ILC2, such as cellular apoptosis and lipid metabolic pathways, as well as the presence of miRNAs, require further investigation.

### 2.5. Th17 Cells

#### 2.5.1. Th17 Cells in Virus-Induced Asthma Exacerbations

Th17 cells also play an important part in virus-induced asthma, mediated by the role of IL-17 family cytokines [116,117,118]. In basic experiments in which an OVA mouse model was infected with RSV, IL-17A regulated the airway hyperreactivity [116]. Niwa et al. demonstrated in vitro experiments using normal human bronchial epithelium that the increase in IFN-λ, which plays a protective role in viral infection, was attenuated by the presence of IL-17A [118]. Increased IL-17 is thought to be one of the mechanisms of steroid resistance in asthmatic patients [118], and IL-17 may be a novel therapeutic target for patients with viral-induced asthma who are refractory to treatment.

IL-6 and transforming growth factor (TGF)-β are required for Th17 differentiation. Viral infection, including HRV infection, increases IL-6 levels in the respiratory tract [119,120], which correlates with airway remodeling and the severity of asthma [121,122]. At present, the anti-IL-6 antibody tocilizumab, which is used to treat rheumatoid arthritis, is not indicated for asthma, but basic experiments suggested that IL-6 may be an important marker of asthma [123,124].

TGF-β is an essential factor affecting airway remodeling along with MMP-9, amphiregulin, vascular endothelial growth factor (VEGF), and fibroblast growth factor (FGF) [125]. Repeated RV infection in mice not sensitized to allergens activated TGF-β in lung tissue, but neutralizing TGF-β reduced airway smooth muscle thickening [125]. TGF-β-deficient mice showed an earlier increase in IFN-β in lung tissue compared to that in controls [126]. Collectively, TGF-β was found to contribute strongly to remodeling during virus-induced asthma exacerbations.

One therapeutic candidate targeting Th17 cells is brodalumab, a monoclonal antibody against the IL-17 receptor, which has failed to show efficacy in the clinical trial [127]. Further elucidation of their mechanisms and development of therapeutic agents are required to control the pathogenesis of virus-induced asthma exacerbations.

#### 2.5.2. The Relationship between SIRT1 and Th17 Cells in Virus-Induced Asthma Exacerbations

It was reported that loss of SIRT1 further induced mRNA expression of IL-17 in an animal model of RSV infection [128]. According to their considerations, SIRT1-deficient bone marrow dendritic cells elevated Acetyl CoA carboxylase 1 (ACC1), which is associated with fatty acid synthesis, resulting in the activation of an abnormal metabolic process, which in turn preceded an excessive virus-induced immune response [128]. In other words, SIRT1 may regulate the Th17 immune response from virus-infected dendritic cells by regulating their metabolic pathways [128]. The relationship between SIRT1 and IL-17 has been well studied in other diseases. Previous studies investigated the effect of SIRT1 activation in patients with psoriasis [129,130]. They found a significant histological improvement in the SIRT activator group compared to that in the placebo group, which was attributed to the inhibition of IL-17 and TNF-α [129]. Moreover, resveratrol, a SIRT1 activator, suppressed the expression of CCL6, a chemokine that is essential chemokine for the production of IL-17 [130]. The involvement of Th17 cells was demonstrated in patients with diabetic ophthalmopathy [130,131]. Other studies reported that the SIRT1 activator might inhibit the elevation in serum IL-17 levels, and regulation of IL-17 through SIRT1 probably leads to a reduction in the development of diabetic ophthalmopathy [132]. By applying the proven relationship between SIRT1 and IL-17 in these other diseases to viral-induced asthma, SIRT1 could become a new therapeutic target in the future.

SIRT1 regulated IL-6 expression in the ovalbumin-induced asthma mouse model [17,133,134]. These studies confirmed that the activation of the PI3K-Akt pathway led to an increase in IL-6, and this response was attenuated by SIRT1 inhibitors [133,134]. Ichikawa et al. showed that a SIRT1 activator suppressed IL-6 and TNF-α production by splenocytes in ovalbumin-challenged mice [17]. These indicate that the Akt-SIRT1 signaling is a crucial pathway to the control of IL-6. Interestingly, another study reported that metformin, a pharmacotherapeutic agent used to treat diabetes, reduced IL-6, IL-17, IL-1β, and TNF-α levels in a mouse model of acute respiratory distress syndrome [135]. One possibility was that the low expression of miRNA138 might suppress the mitogen-activated protein kinase (MAPK) pathway upstream of SIRT1 [135]. Moreover, metformin, like resveratrol, was reported to inhibit HIF-1α expression, but through a different pathway [136]. In clinical practice, oral metformin reduced asthma-related hospitalizations and asthma exacerbations [137]. Metformin may be one of the treatment options in patients with virus-induced asthma exacerbations by regulating Th17 cells.

The relationship between SIRT1 and TGF-β has often been studied in idiopathic pulmonary fibrosis (IPF). Zeng et al. used a bleomycin-induced mouse model to explore the role of SIRT1 in pulmonary fibrosis [138]. They demonstrated that activation of SIRT1 by resveratrol and SRT1720 inhibited myofibroblast differentiation induced by TGF-β1. Overexpression of SIRT6, a member of the SIRT family, reduced E-cadherin, a marker of EMT, through the TGF-β pathway [139]. This result might involve p21, a protein that regulates cell cycle progression [140]. Another study showed that resveratrol inhibits airway remodeling through transforming growth factor (TGF)-β1/Smad signaling [141]. The relationship between SIRT1 and TGF-β, which was identified in IPF, has potential applications in virus-induced asthma exacerbations, but the mechanisms of this relationship are still unclear.

Therefore, the findings of these studies suggest that future treatments against cellular senescence, especially SIRT1, may regulate Th17 airway inflammation. We believe that metformin is an attractive treatment option because it is already used in many patients with few adverse events, but further research is needed to determine whether it is useful in patients with virus-induced asthma exacerbations who do not have diabetes mellitus.

### 2.6. Viral Protein

#### 2.6.1. Viral Protein in Virus-Induced Asthma Exacerbations

Protein acetylation plays a crucial role in host response to viral infection. Histone deacetylases (HDAC) are enzymes that define chromatin structure and are closely associated with chronic respiratory diseases [142]. Adenovirus infection resulted in reduced activation of HDAC in OVA sensitization mice [143]. Experiments using human blood samples also showed reduced activation of HDAC in asthmatics compared to that in healthy subjects [144].

NF-E2-related factor2 (Nrf2) is a transcription factor that has a protective effect on cells from oxidative stress caused by reactive oxygen species. RSV, which frequently causes asthma exacerbations, induces deacetylation of Nrf2 [145]. In vitro models of rhinovirus infection showed that inhibition of S-nitrosoglutathione reductase was reported to increase SQSTM1, an Nrf2-dependent gene, and suppress viral growth, with an effect on airway hypersensitivity [146].

Although these viral proteins are important factors in the pathogenesis of virus-induced asthma exacerbations, there remain unanswered questions concerning established treatment.

#### 2.6.2. The Relationship between SIRT1 and Viral Proteins in Virus-Induced Asthma Exacerbations

Theophylline, a drug for asthma and COPD, activates HDAC and exerts an anti-inflammatory effect. When the activity of HDAC was investigated in LPS-stimulated macrophages, HDACs were activated by the combination of theophylline compared to dexamethasone alone [147]. In a clinical trial examining the effects of low volume theophylline, the addition of theophylline to inhaled steroids improved respiratory function [148]. In patients with acute exacerbations of COPD, theophylline also improved HDAC activity during the stable phase [149]. Interestingly, steroid resistance correlated with a decrease in SIRT, and a combination of steroid, theophylline, curcumin, or resveratrol treatment resulted in an increase in SIRT1 as well as an increase in glucocorticoid receptors [150]. Doxophylline, which is considered to have fewer side effects than conventional theophylline preparations, improved the protein expression of SIRT1, which was reduced by LPS stimulation [151]. As a treatment option for targeting SIRT1, theophylline is very effective for patients with inadequate response to steroids.

SIRT1 regulates the expression of various genes by deacetylating histones and transcription factors, such as NF-ĸB, STAT1, and STAT3 [152,153,154,155]. SIRT1 deacetylates induced NF-ĸB activation in monocytes [152]. SIRT1 inhibits growth hormone-stimulated STAT3 activation in mouse embryonic cells via the deacetylation of STAT3 [153]. Treatment with SIRT1 agonists deacetylates STAT3, which inhibits T-cell differentiation into Th7 and Th17 cells [154]. A recent study reported that the activation of SIRT1 by viral infection further affected STAT1 activation [155]. These studies indicated that transcription factors, including STAT1 and STAT3, may be potential biomarkers in virus-induced asthma with SIRT1-targeted therapy.

SIRT1 also regulates antioxidant genes, which are important antiaging genes, via deacetylation of FOXO3 and Nrf2. Resveratrol increased the expression of Nrf2 in obese rat models and paraquat-induced lung injury mouse models [156,157]. *Apios americana* Medikus is an edible tuberous legume native to Eastern North America. Chu and colleagues showed that *Apios americana* Medikus leaf extract increased the expression of Nrf2 in mouse macrophages stimulated with LPS [158]. At present, there is no specific treatment for Nrf2, and further studies are needed. Roflumilast, a selective inhibitor of phosphodiesterase 4 (PDE4), was reported to reduce exacerbations and hospitalization rates as a treatment for COPD [159]. Roflumilast is highlighted by the fact that it increased the SIRT1 expression along with Nrf2 expression in COPD patients [160]. In vitro, RSV infection increased the Nrf2 expression and its increase was attenuated by roflumilast [161]. The proinflammatory cytokines IL-6, IL-8, and TNF-α were also reduced in a capacity-dependent manner. These results suggest that Nrf2 may be a potential therapeutic target for viral asthma exacerbations.

Middle East Respiratory Syndrome coronavirus (MERS-CoV) pp1ab protein is potentially regulated by SIRT1 [162]. SARS-CoV-2 might have a similar motif, but further study is required. SIRT1 and other sirtuins were also reported to have antiviral roles against several DNA and RNA viruses, including HCMV, HSV-1, adenovirus, and influenza A [163]. Although resveratrol was found to inhibit HRV replication in nasal epithelial cells [164], it is unclear whether this is due to intercellular adhesion molecule-1 (ICAM-1) as the HRV receptor, or reduction or deacetylation on potential viral protein acetylation, as the acetylation of HRV or RSV proteins remains unclear. Caspase 3 cleavage levels, indicators of apoptosis, are known to elevate in MERS-CoV infection [165], and resveratrol was found to reduce caspase 3 cleavage levels with less cytotoxicity [166]. Activation of SIRT1 may reduce apoptosis through deacetylation of viral proteins. In another report using airway epithelial cells, Kim and colleagues examined the relationship between cellular senescence and the replication efficiency of influenza virus [167]. Senescent cells infected with influenza virus had reduced expression of IFN-β, which plays an essential role in the immune response compared to nonsenescent cells. They also examined whether SIRT1, an essential factor in cellular senescence, affected viral replication. Interestingly, SIRT1-knockdown cells showed enhanced expression of proteins associated with influenza virus and reduced cell viability [167].

Collectively, with further research, therapies targeting SIRT1 may control asthma exacerbations through acetylation of the viral protein.

## 3. Conclusions

We reviewed the cellular interactions between SIRT1 and inflammatory cells involved in virus-induced asthma exacerbations (Table 1). Viral infection is a common health problem for children and adults with asthma. Especially with the recent rampant COVID-19 pandemic, treating and preventing viral infections is becoming an area of focus. While the role of various respiratory viruses in inducing the exacerbation of asthma is well established, the pathophysiological mechanisms underlying virus-induced asthma exacerbation and its treatment remain controversial. Existing drugs such as ICS and biologics are among the best treatment options for asthma exacerbation triggered by viral infection, but they may be partially ineffective due to unknown mechanisms. The current guidelines do not consider the potential for therapeutic agents related to SIRT1. The difference in the role of SIRT1 in asthma between adults and children is not clear.

However, as mentioned above, SIRT1-related signaling pathways are closely related to airway inflammation in asthma. A better understanding of the SIRT1 mechanism may aid in the prevention and treatment of virus-induced asthma exacerbations (Figure 2). The development of therapies targeting SIRT1 could be a boon for patients with virus-induced asthma. Further research is needed to clarify the relationship between known drugs and SIRT1 and to explore the development of new drugs related to SIRT1.

## Figures and Tables

**Figure 1 jcm-09-02623-f001:**
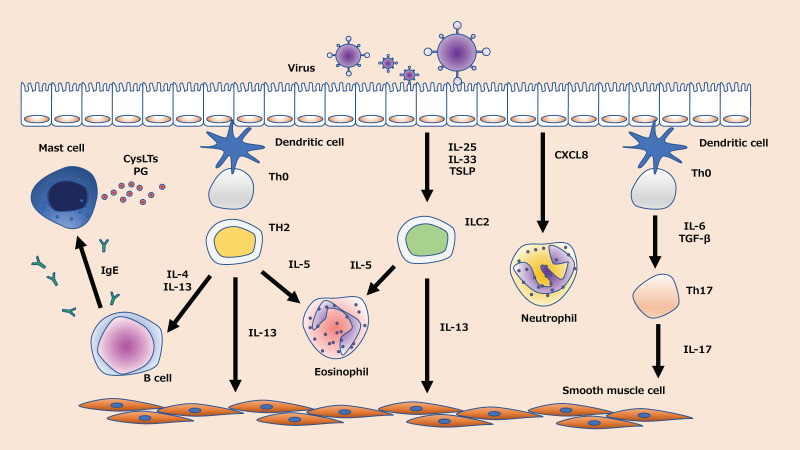
Immunological mechanism of virus-induced asthma exacerbations. When a virus infects the airway epithelium, airway inflammation is induced by multiple pathways. There are two types of airway inflammation: Type 2 airway inflammation involving IL-4, IL-5, IL-13, IgE, and ILC2, and non-Type 2 airway inflammation involving CXCL8 and Th17. CysLTs: cysteinyl leukotrienes. IL: interleukin; ILC2: group 2 innate lymphoid cell; PG: prostaglandin; TBG-β: transforming growth factor-β; TSLP: thymic stromal lymphopoietin.

**Figure 2 jcm-09-02623-f002:**
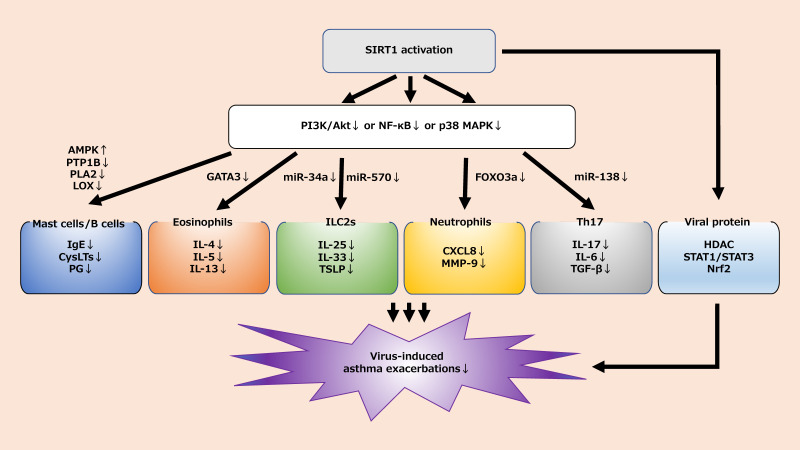
SIRT1-related signaling to virus-induced asthma exacerbations. Activation of SIRT1 promotes the activation of transcription factors, enzyme activity, and viral proteins through PI3K/Akt, NF-κB, and p38 MAPK signaling. As a result, proinflammatory cytokines and chemokines related to mast cells/B cells, eosinophils, ILC2, neutrophils, and Th17 are reduced. Finally, it may lead to the treatment or prevention of virus-induced asthma exacerbations. AMPK; adenosine monophosphate-activated protein kinase: CXCL8; C-X-C motif ligand 8: FOXO3a; forkhead box class O 3a: GATA3; GATA binding protein 3; HDAC; histone deacetylase: IL; interleukin: ILC2; group 2 innate lymphoid cells: LOX; lipoxygenase: MAPK; mitogen-activated protein kinase: miR; microRNA: MMP-9; matrix metalloproteinase-9: Nrf2; NF-E2-related factor 2: PI3K; phosphoinositide 3-kinase: PLA2; phospholipaseA2: PTP1B; protein-tyrosine phosphatase 1B: SIRT1; sirtuin1: STAT; signal transducers and activators of transcription: TGF-β; transforming growth factor (TGF)-β: TSLP; thymic stromal lymphopoietin.

**Table 1 jcm-09-02623-t001:** Relationship between target cells and SIRT1.

Target Cells	Effect of SIRT1 Activation	References (Model)
Neutrophils	CXCL8 ↓	[54,55,56,57,58,59,60] (in vitro)
	MMP-9 ↓	[53,61] (animal)
Eosinophils	IL-4, IL-5, IL-13 ↓	[70,71,72] (in vitro)
	GATA3 ↓	[73] (animal)
Mast cells and B cells	IgE ↓	[92,94] (animal)
		[93] (in vitro)
	LTC_4_, PG ↓	[96,97,98,99] (animal)
	Degranulation ↓	[95] (animal)
ILC2	Il-25, IL-33, TSLP ↓	[109] (in vitro)
	epithelial damage ↓	[110,111] (animal)
	miRNA34a, miRNA570 ↓	[114,115] (in vitro)
Th17 cells		[128] (animal)
	IL-17 ↓	[129,130,131,132] (clinical trial)
		[17,133,134,135,136] (animal)
	IL-6 ↓	[139,140] (in vitro)
	TGF-β↓	[138,141] (animal)
Viral protein	HDAC activity ↑	[149] (clinical trial)
	steroid resistance ↓	[150] (in vitro)
	NF-κB↓	[152] (in vitro)
	STAT1, STAT3 ↓	[154,155] (in vitro)
		[153] (animal)
	Nrf2 ↓	[161] (in vitro)
		[156,157,158] (animal)
		[159,160] (clinical trial)
	Viral replication ↓	[162,163,164,165,166,167] (in vitro)

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
