# Peer review of "Virus-Induced Asthma Exacerbations: SIRT1 Targeted Approach"

_jcm, 2020, doi:10.3390/jcm9082623_

Round 1

Reviewer 1 Report

The manuscript entitled «Virus-induced asthma exacerbations: SIRT1 targeted approach” discusses the role of SIRT1 in inflammatory cells and its role in virus-induced asthma exacerbations. It is a very interesting paper and it might provide insights in future research and clinical trials related with SIRT1 in virus-induced asthma exacerbation. To assure highest quality of the manuscript several comments should be taken into consideration:

Major comments:

  1. General comment: the authors could care take of the most recent citations and in some cases the original paper rather the review should be additionally acknowledged.

  1. In the introduction part, the original papers from the labs of prof. Sebastian Johnston as well as Jim Gern about rhinoviruses should be cited. RSv should be also mentioned in more detail.

  1. A table summarizing relationship between SIRT1 and discussed cell population should be added to the manuscript. It would be useful for the reader to clearly state the source of the evidence (in vitro model vs animal studies vs clinical trials)

  1. Line 51-51. I would suggest to reference the following papers about the different endotypes and phenotypes of asthma in humans and in laboratory animals: PMID: 30855278, PMID: 28987809, PMID: 30267575

  1. Viruses and respiratory viral infections are closely connected to the full dysbiosis of the lung and the disturbance of airway microbiome in different asthma phenotypes and endotypes: PMID: 32075727. It would be beneficial to add a paragraph about it.

  1. Figure 1. Please add to the figure and to the text also the parts about lipid mediators involved in pathology of viral exacerbations of asthma eg. PMID: 23827684, PMID: 32279330, PMID: 27826094. Also, the title of the figure should be rather immunological mechanisms of virus-induced asthma exacerbations. Molecular mechanisms are not really indicated.

  1. Neutrophils are far more abundant in the viral exacerbations than eosinophils, and they are more problematic clinically as the Authors correctly designated, so I would suggest to change the order of the paragraphs.

  1. Paragraph 2.2. The antiviral effect of omalizumab is not through the B cells, but through an effective blockage of IgE. Even if the mechanism is not clear, it would be of interest to discuss it in more detail, especially in regard to SIRT1.

  1. It would be beneficial for the paper if not only the OVA-models were mentioned, but also HDM-induced, as more relevant to human airway inflammation.

  1. I would suggest to add additional paragraph mentioning the possibly unwanted effects of activation of SIRT1, so as it presents the broad scope and a ward of caution.

  1. Page 3 Line 108-109: “They believed that the elevated serum SIRT1 levels were due to the release of SIRT1 from the tissues following airway inflammation.” More convincing supporting results would be better. As listed in Line 101-108, the serum higher SIRT1 levels and worse FEV1/FVC results were from asthmatic patients while decreased level of SIRT1 and related IL were from mouse lung tissue and mouse BALF respectively.

  1. Page 4 Line 132: “IgE is…, which is mainly produced by mast cells and basophils.” Basophils do not produce IgE.

13 Page 4 Line 151-153: “In other words, it may increase IFN-γ against viral infections by activating SIRT1, which may be protective in asthma patients.” Theoretically, based on listed studies, this would be hypothesis rather than observed phenomenon.

  1. Page 4 Line 163: “IgE-mediated airway inflammation and airway remodeling.” No supporting information were demonstrated formerly for airway remodeling.

  1. Is there any evidence regarding SIRT-1-mediated protective effects of diet modification and calorie restriction in asthmatic patients?

Minor comments:

  1. Abstract: I would not suggest to use the word of endo-phenotypes of asthma. It should be endotypes and phenotypes of asthma
  2. Line 105: adjective describing direction of IL-5 and IL-13 expression change in the BALF is missing
  3. Line 213: IFNl2/3 should be INFL2/3
  4. Line 334: Treatment should not start with capital letter
  5. Line 385: Double spacebar should be removed

Author Response

Point 1. General comment: the authors could care take of the most recent citations and in some cases the original paper rather the review should be additionally acknowledged.

Response 1. We appreciate the reviewer’s comments. We have reviewed some references again and cited the more recent original article.

Point 2. In the introduction part, the original papers from the labs of prof. Sebastian Johnston as well as Jim Gern about rhinoviruses should be cited. RSv should be also mentioned in more detail.

Response 2. Thank you for your valuable feedback. We have cited the articles by the lab from Prof. Sebastian Johnston and Prof. Jim Gern for a detailed description of rhinovirus and RS virus infection in introduction part (Page 2 Line 52-63).

Point 3. A table summarizing relationship between SIRT1 and discussed cell population should be added to the manuscript. It would be useful for the reader to clearly state the source of the evidence (in vitro model vs animal studies vs clinical trials)

Response 3. Thank you for the suggestion. We have created a summary table on SIRT1 and inflammatory cells (Table 1). 

Point 4. Line 51-51. I would suggest to reference the following papers about the different endotypes and phenotypes of asthma in humans and in laboratory animals: PMID: 30855278, PMID: 28987809, PMID: 30267575

Response 4. Thank you for your very helpful feedback. We have made some corrections to the manuscript in accordance with the references you pointed out (Page 2 Line 64-67)

Point 5. Viruses and respiratory viral infections are closely connected to the full dysbiosis of the lung and the disturbance of airway microbiome in different asthma phenotypes and endotypes: PMID: 32075727. It would be beneficial to add a paragraph about it.

Response 5. We appreciate the reviewer’s comments. We have added the description of the microbiome you pointed (Page 2 Line 67-69).

Point 6. Figure 1. Please add to the figure and to the text also the parts about lipid mediators involved in pathology of viral exacerbations of asthma eg. PMID: 23827684, PMID: 32279330, PMID: 27826094. Also, the title of the figure should be rather immunological mechanisms of virus-induced asthma exacerbations. Molecular mechanisms are not really indicated.

Response 6. Thank you for your precise suggestion. We have discussed lipid mediators, with a focus on leukotrienes, and revised the manuscript and Figure 1 (Page3, Line 96, Page6 69, Page 6 Line 237-246, Page 6 Line 272-Page7 Line 282).

Point 7. Neutrophils are far more abundant in the viral exacerbations than eosinophils, and they are more problematic clinically as the Authors correctly designated, so I would suggest to change the order of the paragraphs.

Response 7. Thank you for the appropriate suggestion. As you point out, neutrophilic airway inflammation is a more important pathology in virus-induced asthma exacerbations. We have rearranged the order of the paragraphs (Page 3 Line 119-Page 4, Line 168).

Point 8. Paragraph 2.2. The antiviral effect of omalizumab is not through the B cells, but through an effective blockage of IgE. Even if the mechanism is not clear, it would be of interest to discuss it in more detail, especially in regard to SIRT1.

Response 8. Thank you for your valuable feedback. Certainly, omalizumab exerts its effect by blocking IgE, and we have added a description of the mechanism associated with SIRT1 (Page 6 Line 265-271).

Point 9. It would be beneficial for the paper if not only the OVA-models were mentioned, but also HDM-induced, as more relevant to human airway inflammation.

Response 9. To the best of our knowledge, the overwhelming majority of experimental models that examined the effects of SIRT1 were OVA-induced mouse models. However, as you point out, HDM-induced mouse models are an important factor in understanding asthma pathogenesis. Based on reference 110, we have added a description to the manuscript (Page 7 Line 304-306).

Point 10. I would suggest to add additional paragraph mentioning the possibly unwanted effects of activation of SIRT1, so as it presents the broad scope and a ward of caution.

Response 10. Thank you for your suggestion. It is true that activating SIRT1 is not all good. Based on previous reports, we have created a new paragraph on SIRT1, including unwanted effects of activating SIRT1 (Page3 Line 104-118).

Point 11. Page 3 Line 108-109: “They believed that the elevated serum SIRT1 levels were due to the release of SIRT1 from the tissues following airway inflammation.” More convincing supporting results would be better. As listed in Line 101-108, the serum higher SIRT1 levels and worse FEV1/FVC results were from asthmatic patients while decreased level of SIRT1 and related IL were from mouse lung tissue and mouse BALF respectively.

Response 11. Thank you for your important comments. As you mentioned, the description is inadequately supported by the evidence, so we have made corrections in the manuscript based on previous studies (Page5 Line 195-201).

Point 12. Page 4 Line 132: “IgE is…, which is mainly produced by mast cells and basophils.” Basophils do not produce IgE.

Response 12. Your comment is correct. We fixed the manuscript “ which is mainly produced by plasma cells” (Page5 Line 223-224).

Point 13. Page 4 Line 151-153: “In other words, it may increase IFN-γ against viral infections by activating SIRT1, which may be protective in asthma patients.” Theoretically, based on listed studies, this would be hypothesis rather than observed phenomenon.

Response 13. We agree with your assessment. Surely this is a hypothetical, so we deleted the unnecessary statement (Page 6 Line 253-255).

Point 14. Page 4 Line 163: “IgE-mediated airway inflammation and airway remodeling.” No supporting information were demonstrated formerly for airway remodeling.

Response 14. We agree with your assessment. As you pointed out, the link between IgE and remodeling was missing in the manuscript. We removed the phrase "airway remodeling" (Page 6 Line 269-271).

Point 15. Is there any evidence regarding SIRT-1-mediated protective effects of diet modification and calorie restriction in asthmatic patients?

Response 15. Thank you for your fascinating point. No previous studies show that diet modification or calorie restriction has a protective benefit via SIRT1 in patients with asthma. We found some relevant literature in mouse models and research on the association between asthma and diet, so we cited them along with a discussion on Lipid mediator (Page 6 Line 272-Page 7 Line 282).

Minor comments:

Point 1. Abstract: I would not suggest to use the word of endo-phenotypes of asthma. It should be endotypes and phenotypes of asthma

Response 1. Thank you point it out. We changed the word according to the phrasing you pointed out (Page1 Line 29).

Point 2. Line 105: adjective describing direction of IL-5 and IL-13 expression change in the BALF is missing

Response 2. We agree with you. We corrected the wording on the part you pointed (Page 5 Line 195-198).

Point 3. Line 213: IFNl2/3 should be INFL2/3

Response 3. Thank you point it out. That was our spelling error, and We fixed it (Page 4 Line 135).

Point 4. Line 334: Treatment should not start with capital letter

Response 4. We agree with you. We consulted with English-native and removed the word. (Page 9 Line 410).

Point 5. Line 385: Double spacebar should be removed

Response 5. Thank you point it out. We removed double spacebar (Page 11 Line 465).

Reviewer 2 Report

What are the roles of SIRT1 on basophils? Endothelial cells? Eosinophil-basophil progenitors? Alveolar macrophages?

Can the authors comment on the differences in SIRT1 involvement in naïve and memory B and T cells?

Is there a difference in SIRT1 involvement in children vs adults?

The authors should include a table with references snapshotting the roles of SIRT1 on all of the affected cell types – this would be a great overview table.

Author Response

Point 1. What are the roles of SIRT1 on basophils? Endothelial cells? Eosinophil-basophil progenitors? Alveolar macrophages?

Response 1. We appreciate reviewer’s comments. Although a pathway to inhibit Syk kinase has been identified as a role for SIRT1 on basophils (K Miura et al. Clin Exp Allergy. 2001;31(11):1732-9.), much is still unknown. However, it was confirmed that SIRT1 inhibits degranulation from basophils, so we have described in the manuscript (Page 6, Line 265).

Point 2. Can the authors comment on the differences in SIRT1 involvement in naïve and memory B and T cells?

Response 2. The involvement of SIRT1 in naïve and memory B and T cells is unclear, and it is difficult for us to describe the relationship in detail. However, We have added some of the points you raised in the manuscript (Page 3 Line 106-109).

Point 3. Is there a difference in SIRT1 involvement in children vs adults?

Response 3. Thank you for this interesting question. The evidence that currently exists is not clear on the differences in the role of SIRT1 in asthma patients between children and adults. We have added this to the manuscript as one of the limitations (Page 11 Line 469).

Point 4. The authors should include a table with references snapshotting the roles of SIRT1 on all of the affected cell types – this would be a great overview table.

Response 4. Thank you for the suggestion. We created a summary table on SIRT1 and inflammatory cells (Table 1). 

Reviewer 3 Report

In the present review, Yosuke Fukuda. et. al. efficiently summarize the recent literature in the field of virus-induced asthma exacerbations and the biology of the histone deacetylase SIRT-1. The literature review is up-to-date and extensive. The manuscript covers an interesting field of research with a potential impact on treatment options for virus-induced asthma exacerbations. Still, there are several issues that need to be addressed to improve the quality of the manuscript.

Major points:

  1. The review can be improved by reorganizing its presentation to emphasize the most important aspects of the SIRT-1 biology and virus-induced exacerbations, and reduce the other, less important and extensively reviewed features, so that a clearer message is presented to the broad audience.
  2. The manuscript would benefit with the inclusion of a paragraph describing the biology of SIRT-1 and its effects on the regulation of gene expression and on metabolic responses. A description of its major activators and inhibitors is also needed, as this will set the stage for the literature that follows wherein the effects of these molecules on immune responses are described.
  3. Some parts throughout the manuscript are repetitive and should be deleted. For example, on page 2, lines 57-58 are repetitive of the lines 46-48. In this respect, a distinct and concise paragraph describing virus induced asthma exacerbations should be added in the introduction. On lines 198-200, the role of ICS is described; however, they are mentioned also on lines 175-177 and this is confusing for the audience. There should be a description of ICS at the first mention.
  4. There should be a separate paragraph describing current therapeutic modalities for virus-induced asthma exacerbations and which are the unmet clinical needs.
  5. Certain sentences of the manuscript do not have a clear meaning and should be corrected. For example, sentence on lines 104-106 and lines 251-253.
  6. The statement on lines 267-269 is wrong and should be corrected. There are studies wherein IL-17 neutralizing antibodies have been used in asthmatic patients (Busse WW, Holgate S, Kerwin E, Chon Y, Feng J, Lin J, Lin SL. Am J Respir Crit Care Med. 2013).
  7. At certain parts in the review, it is mentioned that SIRT-1 acts as an acetylase even though its principal role is deacetylation (lines 226 and 371). This is confusing and should be explained by the authors.
  8. The figure legends should have a detailed description of the mechanisms that are being illustrated.
  9. The manuscript has several grammar errors and should be revised by an English-native.

Author Response

Point 1. The review can be improved by reorganizing its presentation to emphasize the most important aspects of the SIRT-1 biology and virus-induced exacerbations, and reduce the other, less important and extensively reviewed features, so that a clearer message is presented to the broad audience.

Response 1. We agree with you. There are parts of the message that have lost clarity due to the less critical statements. We've reordered the manuscript and modified it around the elements you pointed out below to clarify the message.

Point 2. The manuscript would benefit with the inclusion of a paragraph describing the biology of SIRT-1 and its effects on the regulation of gene expression and on metabolic responses. A description of its major activators and inhibitors is also needed, as this will set the stage for the literature that follows wherein the effects of these molecules on immune responses are described.

Response 2. Thank you for your suggestion. We created a new paragraph on SIRT1, including biology, and activator/inhibitor of SIRT1 (Page3 Line 104-118).

Point 3. Some parts throughout the manuscript are repetitive and should be deleted. For example, on page 2, lines 57-58 are repetitive of the lines 46-48. In this respect, a distinct and concise paragraph describing virus induced asthma exacerbations should be added in the introduction. On lines 198-200, the role of ICS is described; however, they are mentioned also on lines 175-177 and this is confusing for the audience. There should be a description of ICS at the first mention.

Response 3. Thank you for making a very important point. It is true that some parts of the text are repetitive and have the potential to confuse readers. We have made revisions to the text (Page 2 Line 45-49, Page 7 Line 297-299).

Point 4. There should be a separate paragraph describing current therapeutic modalities for virus-induced asthma exacerbations and which are the unmet clinical needs.

Response 4. Thank you for your suggestion. We have added a description of current treatment strategies for virus-induced asthma exacerbations based on previous reports (Page 2 Line 73-78).

Point 5. Certain sentences of the manuscript do not have a clear meaning and should be corrected. For example, sentence on lines 104-106 and lines 251-253.

Response 5. We agree with you. We have reviewed the description and made corrections to the manuscript (Page 5 Line 195-198, Page 8 Line 323-326).

Point 6. The statement on lines 267-269 is wrong and should be corrected. There are studies wherein IL-17 neutralizing antibodies have been used in asthmatic patients (Busse WW, Holgate S, Kerwin E, Chon Y, Feng J, Lin J, Lin SL. Am J Respir Crit Care Med. 2013).

Response 6. Your comment is correct. I have added a new description based on the references you listed (Page 8 Line 340-342).

Point 7. At certain parts in the review, it is mentioned that SIRT-1 acts as an acetylase even though its principal role is deacetylation (lines 226 and 371). This is confusing and should be explained by the authors.

Response 7. Thank you for pointing that out. It was indeed a confusing statement and I have corrected that part (Page 4 Line 148, Page 10 Line 447).

Point 8. The figure legends should have a detailed description of the mechanisms that are being illustrated.

Response 8. We agree with you. We have added figure legend in detail (Figure 1 and 2). 

Point 9. The manuscript has several grammar errors and should be revised by an English-native.

Response 9. Thank you for the suggestion. We had the manuscript proofread again by an English-native, and noted it in the acknowledge section.

Round 2

Reviewer 1 Report

Thank you for implementing my comments and improving your manuscript.

Reviewer 3 Report

The manuscript has been substantially improved  by the authors and is suitable for publication.

This manuscript is a resubmission of an earlier submission. The following is a list of the peer review reports and author responses from that submission.